# Dynamical Distance Learning for Semi-Supervised and Unsupervised Skill Discovery

**Kristian Hartikainen***
University of California, Berkeley
University of Oxford

**Xinyang Geng**
University of California, Berkeley

**Tuomas Haarnoja**†
University of California, Berkeley
Google DeepMind

**Sergey Levine**†
University of California, Berkeley

## Abstract

Reinforcement learning requires manual specification of a reward function to learn a task. While in principle this reward function only needs to specify the task goal, in practice reinforcement learning can be very time-consuming or even infeasible unless the reward function is shaped so as to provide a smooth gradient towards a successful outcome. This shaping is difficult to specify by hand, particularly when the task is learned from raw observations, such as images. In this paper, we study how we can automatically learn dynamical distances: a measure of the expected number of time steps to reach a given goal state from any other state. These dynamical distances can be used to provide well-shaped reward functions for reaching new goals, making it possible to learn complex tasks efficiently. We show that dynamical distances can be used in a semi-supervised regime, where unsupervised interaction with the environment is used to learn the dynamical distances, while a small amount of preference supervision is used to determine the task goal, without any manually engineered reward function or goal examples. We evaluate our method both on a real-world robot and in simulation. We show that our method can learn to turn a valve with a real-world 9-DoF hand, using raw image observations and just ten preference labels, without any other supervision. Videos of the learned skills can be found on the project website: https://sites.google.com/view/dynamical-distance-learning.

## 1 Introduction

The manual design of reward functions represents a major barrier to the adoption of reinforcement learning (RL), particularly in robotics, where vision-based policies can be learned end-to-end (Levine et al., 2016; Haarnoja et al., 2018c), but still require reward functions that themselves might need visual detectors to be designed by hand (Singh et al., 2019). While in principle the reward only needs to specify the goal of the task, in practice RL can be exceptionally time-consuming or even infeasible unless the reward function is shaped so as to provide a smooth gradient towards a successful outcome. Prior work tackles such situations with dedicated exploration methods (Houthooft et al., 2016; Osband et al., 2016; Andrychowicz et al., 2017), or by using large amounts of random exploration (Mnih et al., 2015), which is feasible in simulation but infeasible for real-world robotic learning. It is also common to employ heuristic shaping, such as the Cartesian distance to a goal for an object relocation task (Mahmood et al., 2018; Haarnoja et al., 2018a). However, this kind of shaping is brittle and requires manual insight, and is often impossible when ground truth state observations are unavailable, such as when learning from image observations.

---

*Correspondence to kristian.hartikainen@cs.ox.ac.uk
†Equal advising.

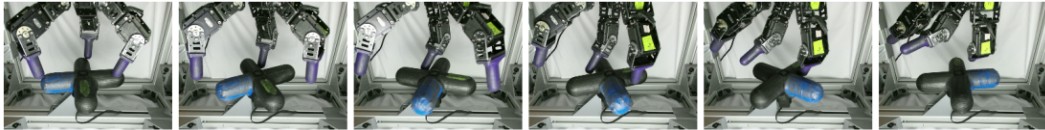

Figure 1: We present a dynamical distance learning (DDL) method that can learn a 9-DoF real-world dexterous manipulation task directly from raw image observations. DDL does not assume access to the true reward function and solves the 180 degree valve-rotation task in 8 hours by relying only on 10 human-provided preference labels.

In this paper, we aim to address these challenges by introducing dynamical distance learning (DDL), a general method for learning distance functions that can provide effective shaping for goal-reaching tasks without manual engineering. Instead of imposing heuristic metrics that have no relationship to the system dynamics, we quantify the distance between two states in terms of the number of time steps needed to transition between them. This is a natural choice for dynamical systems, and prior works have explored learning such distances in simple and low-dimensional domains (Kaelbling, 1993). While such distances can be learned using standard model-free reinforcement learning algorithms, such as Q-learning, we show that such methods generally struggle to acquire meaningful distances for more complex systems, particularly with high-dimensional observations such as images. We present a simple method that employs supervised regression to fit dynamical distances, and then uses these distances to provide reward shaping, guide exploration, and discover distinct skills.

The most direct use of DDL is to provide reward shaping for a standard deep RL algorithm, to optimize a policy to reach a given goal state. We can also formulate a semi-supervised skill learning method, where a user expresses preferences over goals, and the agent autonomously collects experience to learn dynamical distances in a self-supervised way. Finally, we can use DDL in a fully unsupervised method, where the most distant states are selected for exploration, resulting in an unsupervised reinforcement learning procedure that discovers difficult skills that reach dynamically distant states from a given start state. All of these applications avoid the need for manually designed reward functions, demonstrations, or user-provided examples, and involve minimal modification to existing deep RL algorithms.

DDL is a simple and scalable approach to learning dynamical distances that can readily accommodate raw image inputs and, as shown in our experiments, substantially outperforms prior methods that learn goal-conditioned policies or distances using approximate dynamic programming techniques, such as Q-learning. We show that using dynamical distances as a reward function in standard reinforcement learning methods results in policies that take the shortest path to a given goal, despite the additional shaping. Empirically, we compare the semi-supervised variant of our method to prior techniques for learning from preferences. We also compare our method to prior methods for unsupervised skill discovery on tasks ranging from 2D navigation to quadrupedal locomotion. Our experimental evaluation demonstrates that DDL can learn complex locomotion skills without any supervision at all, and that the preferences-based version of DDL can learn to turn a valve with a real-world 9-DoF hand, using raw image observations and 10 human-provided preference labels, without any other supervision.

## 2 RELATED WORK

Dynamical distance learning is most closely related to methods that learn goal-conditioned policies or value functions (Schaul et al., 2015; Sutton et al., 2011). Many of these works learn goal-reaching directly via model-free RL, often by using temporal difference updates to learn the distance function as a value function (Kaelbling et al., 1996; Schaul et al., 2015; Andrychowicz et al., 2017; Pong et al., 2018; Nair et al., 2018; Florensa et al., 2019). For example, Kaelbling (1993) learns a goal conditioned Q-function to represent the shortest path between any two states, and Andrychowicz et al. (2017) learns a value function that resembles a distance to goals, under a user-specified low-dimensional goal representation. Unlike these methods, DDL learns policy-conditioned distances with an explicit supervised learning procedure, and then employs these distances to recover a reward function for RL. We experimentally compare to RL-based distance learning methods, and show that

DDL attains substantially better results, especially with complex observations. Another line of prior work uses a learned distance to build a search graph over a set of visited states (Savinov et al., 2018; Eysenbach et al., 2019), which can then be used to plan to reach new states via the shortest path. Our method also learns a distance function separately from the policy, but instead of using it to build a graph, we use it to obtain a reward function for a separate model-free RL algorithm.

The semi-supervised variant of DDL is guided by a small number of preference queries. Prior work has explored several ways to elicit goals from users, such as using outcome examples and a small number of label queries (Singh et al., 2019), or using a large number of relatively cheap preferences (Christiano et al., 2017). The preference queries that our semi-supervised method uses are easy to obtain and, in contrast to prior work (Christiano et al., 2017), we only need a small number of these queries to learn a policy that reliably achieves the user's desired goal. Our method is also well suited for fully unsupervised learning, in which case DDL uses the distance function to propose goals for unsupervised skill discovery. Prior work on unsupervised reinforcement learning has proposed choosing goals based on a variety of unsupervised criteria, typically with the aim of attaining broad state coverage (Nair et al., 2018; Florensa et al., 2018; Eysenbach et al., 2018; Warde-Farley et al., 2018; Pong et al., 2019). Our method instead repeatedly chooses the most distant state as the goal, which produces rapid exploration and quickly discovers relatively complex skills. We provide a comparative evaluation in our experiments.

## 3 PRELIMINARIES

In this work, we study control of systems defined by fully observed Markovian dynamics $p(\mathbf{s}'|\mathbf{s}, \mathbf{a})$ : $\mathcal{S} \times \mathcal{S} \times \mathcal{A} \to \mathbb{R}_{\geq 0}$, where $\mathcal{S}$ and $\mathcal{A}$ are continuous state and action spaces. We aim to learn a stochastic policy $\pi(\mathbf{a}|\mathbf{s}) : \mathcal{A} \times \mathcal{S} \to \mathbb{R}_{\geq 0}$, to reach a goal state $\mathbf{g} \in \mathcal{S}$. We will denote a trajectory with $\tau \triangleq (\mathbf{s}_0, \mathbf{a}_0, ..., \mathbf{s}_T) \sim \rho_\pi$, where $\rho_\pi$ is a the trajectory distribution induced by the policy $\pi$, and $\mathbf{s}_0$ is sampled from an initial state distribution $\rho(\mathbf{s}_0)$. The policy can be optimized using any reinforcement learning algorithm by maximizing

$$\mathcal{L}(\pi) = \mathbb{E}_{\tau \sim \rho_\pi} \left[ \sum_{t=0}^{\infty} \gamma^t r_{\mathbf{g}}(\mathbf{s}_t, \mathbf{a}_t) \right], \tag{1}$$

where $r_{\mathbf{g}} : \mathcal{S} \times \mathcal{A} \to [-R_{\min}, R_{\max}]$ is a bounded reward function and $\gamma \in [0, 1)$ is a discount factor.[1] However, we *do not* assume that we have access to a shaped reward function. In principle, we could set the reward to $r_{\mathbf{g}}(\mathbf{s}, \mathbf{a}) = 0$ if $\mathbf{s} = \mathbf{g}$ and $r_{\mathbf{g}}(\mathbf{s}, \mathbf{a}) = -1$ otherwise to learn a policy to reach the goal in as few time steps as possible. Unfortunately, such a sparse reward signal is extremely hard to optimize, as it does not provide any gradient towards the optimal solution until the goal is actually reached. Instead, in Section 4, we will show that we can efficiently learn to reach goals by making use of a learned dynamical distance function.

## 4 DYNAMICAL DISTANCE LEARNING

The aim of our method is to learn policies that reach goal states. These goal states can be selected either in an unsupervised fashion, to discover complex skills, or selected manually by the user. The learning process alternates between two steps: in the *distance evaluation* step, we learn a policy-specific dynamical distance, which is defined in the following subsection. In the *policy improvement* step, the policy is optimized to reach the desired goal by using the distance function as the negative reward. This process will lead to a sequence of policies and dynamical distance functions that converge to an effective goal-reaching policy. Under certain assumptions, we can prove that this process converges to a policy that minimizes the distance from any state to any goal, as discussed in Appendix B. In this section, we define dynamical distances and describe our dynamical distance learning (DDL) procedure. In Section 5, we will describe the different ways that the goals can be chosen to instantiate our method as a semi-supervised or unsupervised skill learning procedure.

---

[1]In practice, we use soft actor-critic to learn the policy, which uses a related maximum entropy objective (Haarnoja et al., 2018c).

## 4.1 DYNAMICAL DISTANCE FUNCTIONS

The dynamical distance associated with a policy $\pi$, which we write as $d^\pi(\mathbf{s}_i, \mathbf{s}_j)$, is defined as the expected number of time steps it took for $\pi$ to reach a state $\mathbf{s}_j$ from a state $\mathbf{s}_i$, given that the two were visited in the same episode.[2] Mathematically, the distance is defined as:

$$d^\pi(\mathbf{s}, \mathbf{s}') \triangleq \mathbb{E}_{\tau \sim \pi | \mathbf{s}_i = \mathbf{s}, \mathbf{s}_j = \mathbf{s}', \; j \geq i} \left[ \sum_{t=i}^{j-1} \gamma^{t-i} c(\mathbf{s}_t, \mathbf{s}_{t+1}) \right], \tag{2}$$

where $\tau$ is sampled from the conditional distribution of trajectories that passes through first $\mathbf{s}$ and then $\mathbf{s}'$, and where $c$ is some local cost of moving from $\mathbf{s}_i$ to $\mathbf{s}_{i+1}$. For example, in a typical case in the absence of supervision, we can set $c(\mathbf{s}_t, \mathbf{s}_{t+1}) \equiv 1$ analogously to the binary reward function in Equation 1, in which case the sum reduces to $j - i$, and we recover the expected number of time steps to reach $\mathbf{s}'$. In principle, we could also trivially incorporate more complex local costs $c$, for example to include action costs. This modification would be straightforward, though we focus on the simple $c(\mathbf{s}_t, \mathbf{s}_{t+1}) \equiv 1$ in our derivation and experiments. We include the discount factor to extend the definition to infinitely long trajectories, but in practice we set $\gamma = 1$.

## 4.2 DISTANCE EVALUATION

In the distance evaluation step, we learn a distance function $d_\psi^\pi(\mathbf{s}, \mathbf{s}')$, parameterized by $\psi$, to estimate the dynamical distance between pairs of states visited by a given policy $\pi_\phi$, parameterized by $\phi$. We first roll out the policy multiple times to sample trajectories $\tau_k$ of length $T$. The empirical distance between states $\mathbf{s}_i, \mathbf{s}_j \in \tau_k$, where $0 \leq i \leq j \leq T$, is given by $j - i$. Because the trajectories have a finite length, we are effectively ignoring the cases where reaching $\mathbf{s}_j$ from $\mathbf{s}_i$ would take more than $T - i$ steps, biasing this estimate toward zero, but since the bias becomes smaller for shorter distances, we did not find this to be a major limitation in practice. We can now learn the distance function via supervised regression by minimizing

$$\mathcal{L}_d(\psi) = \frac{1}{2} \mathbb{E}_{\substack{\tau \sim \rho_\pi \\ i \sim [0, T] \\ j \sim [i, T]}} \left[ \left( d_\psi^\pi(\mathbf{s}_i, \mathbf{s}_j) - (j - i) \right)^2 \right]. \tag{3}$$

As we will show in our experimental evaluation, this supervised regression approach makes it feasible to learn dynamical distances for complex tasks with raw image observations, something that has proven exceptionally challenging for methods that learn distances via goal-conditioned policies or value functions and rely on temporal difference-style methods. In direct comparisons, we find that such methods generally struggle to learn on the more complex tasks with image observations. On the other hand, a disadvantage of supervised regression is that it requires on-policy experience, potentially leading to poor sample efficiency. However, because we use the distance as an intermediate representation that guides off-policy policy learning, as we will discuss in Section 4.3, we did not find the on-policy updates for the distance to slow down learning. Indeed, our experiments in Section 6.1 show that we can learn a manipulation task on a real robot with roughly the same amount of experience as is necessary when using a well-shaped and hand-tuned reward function.

## 4.3 POLICY IMPROVEMENT

In the policy improvement step, we use $d_\psi^\pi$ to optimize a policy $\pi_\phi$, parameterized by $\phi$, to reach a goal $\mathbf{g}$. In principle, we could optimize the policy by choosing actions that greedily minimize the distance to the goal, which essentially treats negative distances as the values of a value function, and would be equivalent to the policy improvement step in standard policy iteration. However, acting greedily with respect to the dynamical distance defined in Equation 2 would result in a policy that is optimistic with respect to the dynamics.

This is because the dynamical distance is defined as the expected number of time steps *conditioned* on the policy successfully reaching the second state from the first state, and therefore does not account for the case where the second state is not reached successfully. In some cases, this results in pathologically bad value functions. For example, consider the MDP shown on the right,

---

[2]Dynamical distances are not true distance metrics, since they do not in general satisfy triangle inequalities.

where the agent can reach the goal $\mathbf{g}$ using one of two paths. The first path has one intermediate state that leads to the target state with probability $p$, and an absorbing terminal state $\mathbf{s_T}$ with probability $1 - p$. The other path has two intermediate states, but allows the agent to reach the target every time. The optimal dynamical distance will be 2, regardless of the value of $p$, causing the policy to always choose the risky path and potentially miss the target completely.

The definition of dynamical distances in Equation 2 follows directly from how we learn the distance function, by choosing both $\mathbf{s}_i$ and $\mathbf{s}_j$ from the same trajectory. Conditioning on both $\mathbf{s}_i$ and $\mathbf{s}_j$ is needed when the state space is continuous or large, since visiting two states by chance has zero or near-zero probability. We instead propose to use the distance as a negative reward, and apply reinforcement learning to minimize the cumulative distance on the path to the goal:

$$\mathcal{L}_\pi(\phi) = \mathbb{E}_{\tau \sim \rho_\pi} \left[ \sum_{t=0}^{\infty} \gamma^t d_\psi^\pi(\mathbf{s}_t, \mathbf{g}) \right]. \tag{4}$$

This amounts to minimizing the cumulative distance over visited states, and thus taking a risky action becomes unfavourable if it takes the agent to a state that is far from the target at a later time. We further show that, under certain assumption, the policy that optimizes Equation 4 will indeed acquire the correct behavior, as discussed in Appendix A, and will converge to a policy that takes the shortest path to the goal, as we show in Appendix B.

We note that our simulated experiments below are run in deterministic environments and we do not fully understand why cumulative distances work better than greedily minimizing the distances even in those cases. A comparison between these two cases is shown in Section 6.2.

## 4.4 ALGORITHM SUMMARY

The dynamical distance learning (DDL) algorithm is described in Figure 1. Our implementation uses soft actor-critic (SAC) (Haarnoja et al., 2018c) as the policy optimizer, but one could also use any other off-the-shelf algorithm. In each iteration, DDL first samples a trajectory using the current policy, and saves it in a replay pool $\mathcal{D}$. In the second step, DDL updates the distance function by minimizing the loss in Equation 3. The distance function is optimized for a fixed number of $N_d$ stochastic gradient steps. Note that this method requires that we use recent experience from $\mathcal{D}$, so

---

**Algorithm 1** Dynamical Distance Learning

1: **Input:** $\phi, \psi$ ▷ Initial policy and distance parameters
2: **Input:** $\mathcal{D}$ ▷ Empty replay pool
3: **repeat**
4:   $\tau \sim \rho_\pi$, $\mathcal{D} \leftarrow \mathcal{D} \cup \tau$ ▷ Sample a new trajectory
5:   **for** $i = 0$ **to** $N_d$ **do**
6:     $\psi \leftarrow \psi - \lambda_d \hat{\nabla} \mathcal{L}_d(\psi; \pi)$ ▷ Minimize distance loss
7:   **end for**
8:   $\mathbf{g} \leftarrow \text{choose\_goal}(\mathcal{D})$ ▷ Choose goal state
9:   **for** $i = 0$ **to** $N_\pi$ **do**
10:     $\phi \leftarrow \phi - \lambda_\pi \hat{\nabla} \mathcal{L}_\pi(\phi; d, \mathbf{g})$ ▷ Minimize policy loss
11:   **end for**
12: **until** converged

---

as to learn the distance corresponding to the current policy. In the third step, DDL chooses a goal state from the recent experience buffer. We will describe two methods to choose these goal states in Section 5. In the fourth step, DDL updates the policy by taking $N_\pi$ gradient steps to minimize the loss in Equation 4. The implementation of this step depends on the RL algorithm of choice. These steps are then repeated until convergence.

## 5 GOAL PROPOSALS

In the previous section, we discussed how we can utilize a learned distance function to efficiently optimize a goal-reaching policy. However, a learned distance function is only meaningful if evaluated at states from the distribution it has been trained on, suggesting that the goal states should be chosen from the replay pool. Choosing a goal that the policy can already reach might at first appear strange, but it turns out to yield efficient directed exploration, as explained next.

Simple random exploration, such as $\epsilon$-greedy exploration or other strategies that add noise to the actions, can effectively cover states that are close to the starting state, in terms of dynamical distance. However, when high-reward states or goal states are far away from the start state, such naïve strategies are unlikely to reach them. From this observation, we can devise a simple and effective exploration strategy that leverages the learned dynamical distances: we first use the policy to reach a known goal as quickly as possible and then explore the vicinity of that goal. This way more time is

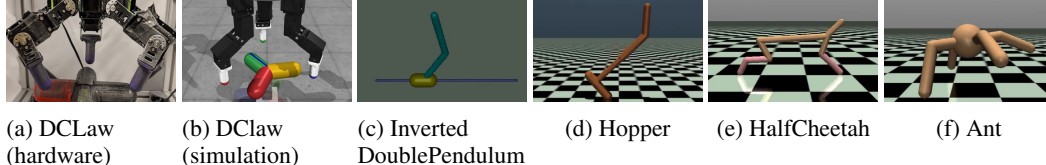

| (a) DCLaw (hardware) | (b) DCLaw (simulation) | (c) Inverted DoublePendulum | (d) Hopper | (e) HalfCheetah | (f) Ant |

Figure 2: We evaluate our method both in simulation and on a real-world robot. We show that our method can learn to turn a valve with a real-world 9-DoF hand (a), and run ablations in the simulated version of the same task (b). We also demonstrate that our method can learn pole balancing (c) and locomotion (d, e, f) skills in simulation.

left to randomly explore states far from the initial state and this way likely discovering useful states. We propose two different strategies for choosing the goals below.

## 5.1 Semi-Supervised Learning from Preferences

DDL can be used to learn to reach specific goals elicited from a user. The simplest way to do this is for a user to provide the goal state directly, either by specifying the full state, or selecting the state manually from the replay pool. However, we can also provide a more convenient way to elicit the desired state with preference queries. In this setting, the user is repeatedly presented with a small slate of candidate states from the replay pool, and asked to select the one that they prefer most. In practice, we present the user with a visualization of the final state in several of the most recent episodes, and the user selects the one that they consider closest to their desired goal.

For example, if the user wishes to train a legged robot to walk forward, they might pick the state where the robot has progressed the largest distance in the desired direction. The required user effort in selecting these states is minimal, and most of the agent's experience is still unsupervised, simply using the latest user-chosen state as the goal. In our experiments, we show that this semi-supervised learning procedure, which we call dynamical distance learning from preferences (DDLfP) can learn to rotate a valve with real-world hand from just ten queries, and can learn simulated locomotion tasks using 100 simulated queries.

## 5.2 Unsupervised Exploration and Skill Acquisition

We can also use DDL to efficiently acquire complex behaviors, such as locomotion skills, in a completely unsupervised fashion. From the observation that many high-reward states are far away from the start state, we can devise a simple and effective exploration strategy that leverages our learned dynamical distances: we can simply select goals that are far from the initial state according to their estimated dynamical distance. We call this variant of our method "dynamical distance learning - unsupervised" (DDLUS).

Intuitively, this method causes the agent to explore the "frontier" of hard-to-reach states, either discovering shorter paths for reaching them and thus making them no longer be on the frontier, or else finding new states further on the fringe through additive random exploration. In practice, we find that this allows the agent to quickly explore distant states in a directed fashion. In Section 6, we show that, by setting $\text{choose\_goal}(\mathcal{D}) \equiv \arg\max_{\mathbf{g} \in \mathcal{D}} d_\psi^\pi(\mathbf{s}_0, \mathbf{g})$, where $\mathbf{s}_0$ is the initial state, we can acquire effective running gaits and pole balancing skills in a variety of simulated settings. While this approach is not guaranteed to discover interesting and useful skills in general, we find that, on a variety of commonly used benchmark tasks, this approach to unsupervised goal selection actually discovers behaviors that perform better with respect to the (unknown) task reward than previously proposed unsupervised reinforcement learning objectives.

## 6 Experiments

Our experimental evaluation aims to study the following empirical questions: **(1)** Does supervised regression provide a good estimator of the true dynamical distance? **(2)** Is DDL applicable to real-

world, vision-based robotic control tasks? **(3)** Does DDL provide an efficient method of learning skills a) from user-provided preferences, and b) completely unsupervised?

We evaluate our method both in the real world and in simulation on a set of state- and vision-based continuous control tasks. We consider a 9-DoF real-world dexterous manipulation task and 4 standard OpenAI Gym tasks (Hopper-v3, HalfCheetah-v3, Ant-v3, and InvertedDoublePendulum-v2). For all of the tasks, we parameterize our distance function as a neural network, and use soft actor-critic (SAC) (Haarnoja et al., 2018b) with the default hyperparameters to learn the policy. For state-based tasks, we use feed-forward neural networks and for the vision-based tasks we add a convolutional preprocessing network before these fully connected layers. The image observation for all the vision-based tasks are 3072 dimensional (32x32 RGB images). Further details are presented in Appendix E.

We study question **(1)** using a simple didactic example involving navigation through a two-dimensional S-shaped maze, which we present in Appendix C. The other two research questions are studied in the following sections.

## 6.1 VISION-BASED REAL-WORLD MANIPULATION FROM HUMAN PREFERENCES

To study the question **(2)**, we apply DDLfP to a real-world vision-based robotic manipulation task. The domain consists of a 9-DoF "DClaw" hand introduced by Ahn et al. (2019), and the manipulation task requires the hand to rotate a valve 180 degrees, as shown in Figure 1. The human operator is queried for a preference every 10K environment steps. Both the vision- and state-based experiments with the real robot use 10 queries during the first 4 hours of an 8-hour training period. Note that, for this and all the subsequent experiments, DDLfP does not have access to the true reward, and must learn entirely from preference queries, which in this case are provided by a human operator.

Figure 3 presents the performance over the course of training. DDLfP uses 10 preference queries to learn the task and its performance is comparable to that of SAC trained with a ground truth shaped reward function. We also

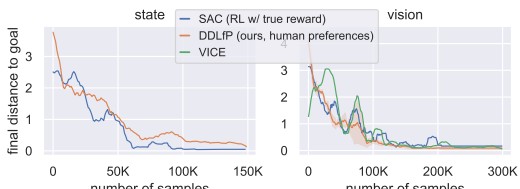

Figure 3: (Left) learning curves for the valve rotation task learned from state. (Right) Same task from vision. The curves correspond to the final distance (measured in radians) of the valve from the target angle during a rollout. Our method (DDLfP, orange) solves the task in 8 hours. Its performance is comparable to that of SAC with true rewards, and VICE with example outcome images. DDLfP only requires 10 preference queries, and learns without true rewards or outcome images. We compare our method in the simulated version of this task in Figure 5.

show a comparison to variational inverse control with events (VICE) (Singh et al., 2019), a recent classifier-based reward specification framework. Instead of preference queries, VICE requires the user to provide examples of the desired goal state at the beginning of training (20 images in this case). For vision-based tasks, VICE involves directly showing images of the desired outcome to the user, which requires physically arranging a scene and taking a picture of it. Preferences, on the other hand, require a user to simply select one state out of a small set, which can be done with a button press and done e.g. remotely, thus often making it substantially less labor-intensive than VICE. As we can see in the experiments, DDLfP achieves similar performance with substantially less operator effort, using only a small number of preference queries. The series of goal preferences queried from the human operator are shown in Appendix D.

## 6.2 ABLATIONS, COMPARISONS, AND ANALYSIS

Next, we analyze design decisions in our method and compare it to prior methods in simulation. First, we replace the cumulative objective in Equation 4 with objective that greedily minimizes the distance function trained with supervised loss. This objective is unable to learn the task from either state or vision observations. Next, we replace the supervised loss in Equation 3 of our DDL method with a temporal difference (TD) Q-learning style update rule that learns dynamical distances with approximate dynamic programming. The results in Figure 5 show that, all else being equal, the TD-based method fails to learn successfully from both low-dimensional state and vision observa-

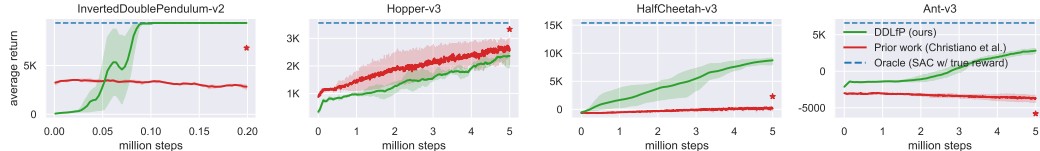

Figure 4: Learning curves for MuJoCo tasks with DDLfP. The y-axis presents the true return of the task. We compare DDLfP to SAC trained directly from the true reward function, which provides an oracle upper bound baseline, and the prior method proposed by Christiano et al. (2017). The prior method uses an on-policy RL algorithm which typically requires more samples than off-policy algorithms, and thus we also plot its final performance after 20M training steps with red star. At the time of the submission, the Ant-v3 run is still in progress and the complete learning curve will be included in the final.

tions. Figure 5 further shows a comparison between using the dynamical distance as the reward in comparison to a reward of -1 for each step until the goal is reached, which corresponds to hindsight experience replay (HER) with goal sampling replaced with preference goals (Andrychowicz et al., 2017). We see that dynamical distances allow the policy to reach the goal when learning both from state and from images, while HER is only successful when learning from low-dimensional states.

These results are corroborated by prior results in the literature that have found that temporal difference learning struggles to capture the true value accurately (Lillicrap et al., 2015; Fujimoto et al., 2018). Note that prior work work does not use the full state as the goal, but rather manually selects a low-dimensional subspace, such as the location of an object, forcing the distance to focus on task-relevant objects (Andrychowicz et al., 2017). Our method learns distances between full image states (3072-dimensional) while HER uses 3-dimensional goals, a difference of two orders of magnitude in dimensionality. This difficulty of learning complex image-based goals is further corroborated in prior work (Pong et al., 2018; Nair et al., 2018; Pong et al., 2019; Warde-Farley et al., 2018).

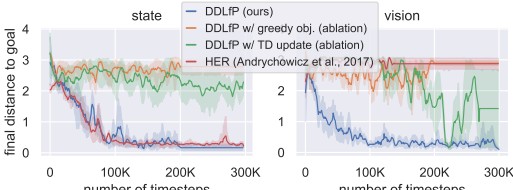

Figure 5: We compare DDL against alternative methods for learning distances on the simulated valve turning task, when learning from the underlying low-dimensional state (left) and from images (right). Dynamical distances used greedily (orange) or learned with TD (green) generally perform poorly. HER (red) can learn from low-dimensional states, but fails to learn from images. Our method, DDLfP (blue) successfully learns the task from either states or images.

Figure 4 presents results for learning from preferences via DDLfP (in green) on a set of continuous control tasks to further study the question **(3,a)**. The plots show the true reward for each method on each task. DDLfP receives only sparse preferences as task-specific supervision, and the preferences in this case are provided synthetically, choosing the state that has progressed the largest distance from the initial state in the desired direction, i.e. the state with largest x-coordinate value. However, this still provides substantially less supervision signal than access to the true reward for all samples. We compare to (Christiano et al., 2017), which also uses preferences for learning skills, but without the use of dynamical distances. The prior method is provided with 750 preference queries over the course of training, while our method uses 100 for all locomotion tasks, and only a single query for the InvertedDoublePendulum-v2, as the initial state and the goal states coincides.[3] Note that Christiano et al. (2017) utilizes an on-policy RL algorithms, which is less efficient than SAC. However, DDLfP outperforms this prior method in terms of *both* final performance and learning speed on all tasks, except for the Hopper-v3 task.

Locomotion tasks like the ones considered here do not fit into DDL framework directly. In this particular case of locomotion tasks, we can fix the issue by considering a case where the ultimate task is to reach a specific goal, i.e. the operator would always choose the goal to be the state closest

---

[3]In our case, one preference query amounts to choosing one of five states, whereas in (Christiano et al., 2017) a query consists always of two state sequences.

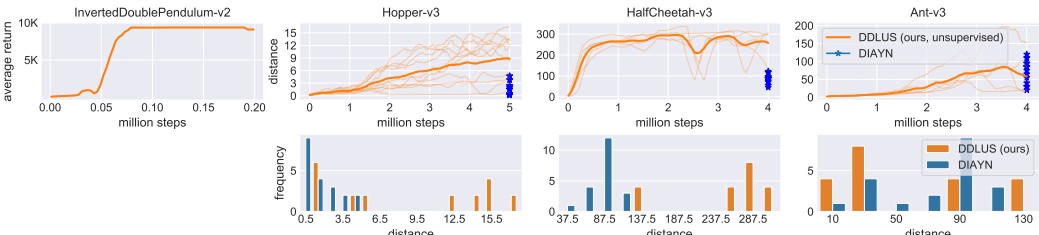

Figure 6: (Top) Learning curves for DDLUS. The y-axis plots the environment return (not accessible during the training) for InvertedDoublePendulum-v3, and the L2-distance travelled from the origin for Hopper-v3, HalfCheetah-v3, and Ant-v3. (Bottom) Frequency histograms of skills learned with DDLUS (blue) and DIAYN (orange) (Eysenbach et al., 2018) across different training runs, evaluated according to the travelled L2-distance from the origin.

to the "ultimate task goal". In that case, we can see the locomotion task to be the limit case where the ultimate goal is as far as possibly reachable within the maximum episode length.

### 6.3 ACQUIRING UNSUPERVISED SKILLS

Finally, we study question **(3,b)** in order to understand how well DDLUS can acquire skills without any supervision. We structure these experiments analogously to the unsupervised skill learning experiments proposed by Eysenbach et al. (2018), and compare to the DIAYN algorithm, another unsupervised skill discovery method, proposed in their prior work. While our method maximizes the complexity of the learned skills by attempting to reach the furthest possible goal, DIAYN maximizes the diversity of learned skills. This of course produces different biases in the skills produced by the two methods. Figure 6 shows both learning curves and histograms of the skills learned in the locomotion tasks with the two methods, evaluated according to how far the simulated robot in each domain travels from the initial state. Our DDLUS method learns skills that travel further than DIAYN, while still providing a variety of different behaviors (e.g., travel in different directions). This experiment aims to provide a direct comparison to the DIAYN algorithm (Eysenbach et al., 2018), though a reasonable criticism is that maximizing dynamical distance is particularly well-suited for the criteria proposed by Eysenbach et al. (2018). We also evaluated DDLUS on the InvertedDoublePendulum-v2 domain, where the task is to balance a pole on a cart. As can be seen from Figure 6, DDLUS can efficiently solve the task without the true reward, as reaching dynamically far states amounts to avoiding failure as far as possible.

## 7 CONCLUSION

We presented dynamical distance learning (DDL), an algorithm for learning dynamical distances that can be used to specify reward functions for goal reaching policies, and support both unsupervised and semi-supervised exploration and skill discovery. Our algorithm uses a simple and stable supervised learning procedure to learn dynamical distances, which are then used to provide a reward function for a standard reinforcement learning method. This makes DDL straightforward to apply even with complex and high-dimensional observations, such as images. By removing the need for manual reward function design and manual reward shaping, our method makes it substantially more practical to employ deep reinforcement learning to acquire skills even with real-world robotic systems. We demonstrate this by learning a valve-turning task with a real-world robotic hand, using 10 preference queries from a human, without any manual reward design or other examples or supervision. One of the main limitations of our current approach is that, although it can be used with an off-policy reinforcement learning algorithm, it requires on-policy data collection for learning the dynamical distances. While the resulting method is still efficient enough to learn directly in the real world, the efficiency of our approach can likely be improved in future work by lifting this limitation. This would not only make learning faster but would also make it possible to pre-train dynamical distances using previously collected experience, potentially making it feasible to scale our method to a multi-task learning setting, where the same dynamical distance function can be used to learn multiple distinct skills.

ACKNOWLEDGMENTS

We thank Vikash Kumar for the DClaw robot design, Nicolas Heess for helpful discussion, and Henry Zhu and Justin Yu for their help on setting up and running the hardware experiments. This research was supported by the Office of Naval Research, the National Science Foundation through IIS-1651843 and IIS-1700696, and Berkeley DeepDrive.

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

# Appendices

## A   CORRECT BEHAVIOR IN THE PATHOLOGICAL MDP

In this appendix we show that the policy that maximizes the objective in Equation 1, with the reward $r_{\mathbf{g}}(\mathbf{s}, \mathbf{a}) = -d^{\pi}(\mathbf{s}, \mathbf{g})$, where $d^{\pi}$ is given by Equation 2, prefers safe actions over risky actions.

Assume that $c(\mathbf{s}_t, \mathbf{s}_{t+1}) = \mathbb{1}_{\mathbf{g}}[\mathbf{s}_t]$ is an indicator function that is 0 if $\mathbf{s}_t$ is a goal state or terminal state and 1 for all the other states. We can now write the definition of $d^{\pi}$ as an infinite sum and substitute $r_{\mathbf{g}}(\mathbf{s}, \mathbf{a}) \leftarrow -d^{\pi}(\mathbf{s}, \mathbf{g})$ in Equation 1:

$$\mathcal{L}(\pi) = -\mathbb{E}_{\tau \sim \pi}\left[\sum_{t=0}^{\infty} \gamma^t \, \mathbb{E}_{\tau' \sim \pi}\left[\sum_{k=0}^{\infty} \gamma^k \mathbb{1}_{\mathbf{g}}[\mathbf{s}_k] \,\middle|\, \mathbf{s}_0' = \mathbf{s}_t, \mathbf{a}_0' = \mathbf{a}_t\right]\right]. \tag{5}$$

The first term ($k = 0$) in the inner sum depends only on $\mathbf{s}_0'$, which is given, and the term can thus be moved outside the inner expectation:

$$\mathcal{L}(\pi) = -\mathbb{E}_{\tau \sim \pi}\left[\sum_{t=0}^{\infty} \gamma^t \mathbb{1}_{\mathbf{g}}[\mathbf{s}_t] + \sum_{t=0}^{\infty} \gamma^t \, \mathbb{E}_{\tau' \sim \pi}\left[\sum_{k=1}^{\infty} \gamma^k \mathbb{1}_{\mathbf{g}}[\mathbf{s}_k'] \,\middle|\, \mathbf{s}_0' = \mathbf{s}_t, \mathbf{a}_0' = \mathbf{a}_t\right]\right]. \tag{6}$$

Next, note that the statistics of the inner expectation over $(\mathbf{s}_1', \mathbf{a}_1')$ are the same as the outer expectation over $(\mathbf{s}_1, \mathbf{a}_1)$, as they are both conditioned on the same $(\mathbf{s}_t, \mathbf{a}_t)$. Thus, we can condition the second expectation directly on $(\mathbf{s}_1', \mathbf{a}_1') = (\mathbf{s}_{t+1}, \mathbf{a}_{t+1})$:

$$\mathcal{L}(\pi) = -\mathbb{E}_{\tau \sim \pi}\left[\sum_{t=0}^{\infty} \gamma^t \mathbb{1}_{\mathbf{g}}[\mathbf{s}_t] + \sum_{t=0}^{\infty} \gamma^t \, \mathbb{E}_{\tau' \sim \pi}\left[\sum_{k=1}^{\infty} \gamma^k \mathbb{1}_{\mathbf{g}}[\mathbf{s}_k'] \,\middle|\, \mathbf{s}_1' = \mathbf{s}_{t+1}, \mathbf{a}_1' = \mathbf{a}_{t+1}\right]\right]. \tag{7}$$

We can now apply the same argument as before and move $\mathbb{1}_{\mathbf{g}}[\mathbf{s}_1']$ outside the inner expectation. Repeating these steps multiple times yields

$$\mathcal{L}(\pi) = -\mathbb{E}_{\tau \sim \pi}\left[\sum_{t=0}^{\infty} \gamma^t \mathbb{1}_{\mathbf{g}}[\mathbf{s}_t] + \sum_{t=0}^{\infty} \gamma^{t+1} \mathbb{1}_{\mathbf{g}}[\mathbf{s}_{t+1}] + \sum_{t=0}^{\infty} \gamma^{t+2} \mathbb{1}_{\mathbf{g}}[\mathbf{s}_{t+2}] + ...\right]$$

$$= -\mathbb{E}_{\tau \sim \pi}\left[\sum_{t=0}^{\infty} \gamma^t (t+1) \mathbb{1}_{\mathbf{g}}[\mathbf{s}_t]\right]. \tag{8}$$

Assuming that the agent always reaches the goal relatively quickly compared to the discount factor, such that $\gamma^t \approx 1$, the trajectories that take longer dominate the loss due to the $(t + 1)$ factor. Therefore, an optimal agent prefers actions that reduce the risk of long, highly suboptimal trajectories, avoiding the pathological behavior discussed in Section 4.3.

## B   POLICY IMPROVEMENT WHEN USING DISTANCE AS REWARD

In this appendix we show that, when we use the negative dynamical distance $-d^{\pi}$ as the reward function in RL, we can learn an optimal policy with respect to the true dynamical distance, leading to policies that optimize the actual number of time steps needed to reach the goal. This result is non-trivial, since the reward function does not at first glance directly optimize for shortest paths. Our proof relies on the assumption that the MDP has deterministic dynamics. However, this assumption holds in all of our experiments, since the MuJoCo benchmark tasks are governed by deterministic dynamics. Under this assumption, DDL will learn policies that take the shortest path to the goal at convergence, despite using the negative dynamical distance as the reward.

Let $d^*(\mathbf{s}, \mathbf{g}) = \min_{\pi} d^{\pi}(\mathbf{s}, \mathbf{g})$ be the optimal distance from state $\mathbf{s}$ to goal state $\mathbf{g}$. Let $\pi'$ be the optimal policy for the reinforcement learning problem with reward $r_{\mathbf{g}}(\mathbf{s}, \mathbf{a}) = -d^{\pi}(\mathbf{s}, \mathbf{g})$. DDL can be viewed as alternating between fitting $d^{\pi}$ to the current policy $\pi$, and learning a new policy $\pi'$ that

is optimal with respect to the reward function given by $-d^{\pi}$.[4] We can now state our main theorem as follows:

**Theorem 1.** *Under deterministic dynamics, for any state $\mathbf{s}$ and $\mathbf{g}$, we have:*

1. $d^{\pi'}(\mathbf{s}, \mathbf{g}) \leq d^{\pi}(\mathbf{s}, \mathbf{g})$.

2. *If $d^{\pi'}(\mathbf{s}, \mathbf{g}) = d^{\pi}(\mathbf{s}, \mathbf{g})$, then $d^{\pi'}(\mathbf{s}, \mathbf{g}) = d^*(\mathbf{s}, \mathbf{g})$.*

*This implies that, when the policy converges, such that $\pi' = \pi$, the policy $\pi'$ achieves the optimal distance to any goal, and therefore is the optimal policy for the shortest path reward function (e.g., the reward function that assigns a reward of $-1$ for any step that does not reach the goal).*

*Proof.*

**Part 1** Without loss of generality, we assume that our policy is deterministic, since the set of optimal policies in an MDP always includes at least one deterministic policy. We also assume that $\mathbf{g}$ is a terminal state and thus $d(\mathbf{g}, \mathbf{g}) = 0$. Let us denote the action of policy $\pi$ on state $\mathbf{s}$ as $\pi(\mathbf{s})$. We start by showing that $d^{\pi'}(\mathbf{s}, \mathbf{g}) \leq d^{\pi}(\mathbf{s}, \mathbf{g})$. We fix a particular goal $\mathbf{g}$. Let $\mathcal{S}_k = \{\mathbf{s}; d^{\pi}(\mathbf{s}, \mathbf{g}) = k\}$ be the set of states that takes $k$ steps under $\pi$ to reach the goal. We show that $d^{\pi'}(\mathbf{s}, \mathbf{g}) \leq d^{\pi}(\mathbf{s}, \mathbf{g}) = k$ for all $\mathbf{s} \in \mathcal{S}_k$ for each $k$ by contradiction.

For $k = 0$, $\mathcal{S}_0 = \{\mathbf{g}\}$ is just the single goal state and $d^{\pi'}(\mathbf{g}, \mathbf{g}) = d^{\pi}(\mathbf{g}, \mathbf{g}) = 0$ by definition. For $k = 1$, for all $\mathbf{s} \in \mathcal{S}_1$, there is an action $\mathbf{a}$ that reaches the goal state as the direct next state. Therefore, the optimized policy $\pi'$ would still take the same action $\mathbf{a}$ on these states and $d^{\pi'}(\mathbf{s}, \mathbf{g}) = 1$.

Now assume that the opposite is true, that $d^{\pi'}(\mathbf{s}, \mathbf{g}) > d^{\pi}(\mathbf{s}, \mathbf{g})$ for some states. Then, there must be a smallest number $K > 1$ and a state $\mathbf{s}_0 \in \mathcal{S}_K$ such that $d^{\pi'}(\mathbf{s}_0, \mathbf{g}) = T > d^{\pi}(\mathbf{s}_0, \mathbf{g}) = K$. Now let us denote the trajectory of states taken by $\pi$ starting from $\mathbf{s}_0$ as $\{\mathbf{s}_0, \mathbf{s}_1, ..., \mathbf{s}_K = g\}$, and the trajectory taken by $\pi'$ as $\{\mathbf{s}'_0 = \mathbf{s}_0, \mathbf{s}'_1, ..., \mathbf{s}'_T = g\}$. Let $\mathcal{L}_{\pi}(\cdot)$ denote the accumulated discounted sum of distance as defined in Equation 4. By our assumption $T > K$, and since $\pi'$ is optimal with respect to the reward $r_{\mathbf{g}}(\mathbf{s}, \mathbf{a}) = -d^{\pi}(\mathbf{s}, \mathbf{g})$, we have

$$\mathcal{L}_{\pi}(\pi') = \sum_{i=0}^{T-1} \gamma^i d^{\pi}(\mathbf{s}'_i, \mathbf{g}) \leq \mathcal{L}_{\pi}(\pi) = \sum_{i=0}^{K-1} \gamma^i d^{\pi}(\mathbf{s}_i, \mathbf{g}) = \sum_{i=0}^{K-1} \gamma^i (K - 1 - i) \qquad (9)$$

Then there must be a time $\hat{t} < K$ such that $d^{\pi}(\mathbf{s}'_{\hat{t}}, \mathbf{g}) < d^{\pi}(\mathbf{s}_{\hat{t}}, \mathbf{g}) = K - 1 - \hat{t}$. Therefore $\mathbf{s}'_{\hat{t}} \in \mathcal{S}_k$ for some $k < K - 1 - \hat{t}$. However, starting from $\mathbf{s}'_{\hat{t}}$, we have $d^{\pi'}(\mathbf{s}'_{\hat{t}}, \mathbf{g}) = T - 1 - \hat{t} > K - 1 - \hat{t} = d^{\pi}(\mathbf{s}'_{\hat{t}}, \mathbf{g})$. Therefore, we reached a contradiction with our assumption that $d^{\pi'}(\mathbf{s}, \mathbf{g}) \leq d^{\pi}(\mathbf{s}, \mathbf{g})$ for all $\mathbf{s}$, $k < K$ such that $\mathbf{s} \in \mathcal{S}_k$. Therefore, $d^{\pi'}(\mathbf{s}, \mathbf{g}) \leq d^{\pi}(\mathbf{s}, \mathbf{g})$ holds for all states.

**Part 2** Now we show the second part: if $d^{\pi}(\mathbf{s}, \mathbf{g}) = d^{\pi'}(\mathbf{s}, \mathbf{g})$, then $d^{\pi}(\mathbf{s}, \mathbf{g}) = d^*(\mathbf{s}, \mathbf{g})$. We prove this with a similar argument, grouping states by distance. Let $\mathcal{S}_k^* = \{\mathbf{s}; d^*(\mathbf{s}, \mathbf{g}) = k\}$ be the set of states that takes $k$ steps under the optimal policy to reach the goal. Note that, for any arbitrary policy $\pi$, we have $d^{\pi}(\mathbf{s}, \mathbf{g}) \geq d^*(\mathbf{s}, \mathbf{g})$ by definition, since $d^*$ is the optimal distance.

Suppose that $d^{\pi}(\mathbf{s}, \mathbf{g}) > d^*(\mathbf{s}, \mathbf{g})$ for some state $\mathbf{s}$. Then there must be a smallest integer $K \geq 0$ such that there exists a state $\mathbf{s}_0 \in \mathcal{S}_K^*$ where $d^{\pi}(\mathbf{s}_0, \mathbf{g}) > d^*(\mathbf{s}_0, \mathbf{g})$. For all $k < K$, we have $d^{\pi}(\mathbf{s}, \mathbf{g}) = d^*(\mathbf{s}, \mathbf{g})$ for all $\mathbf{s} \in \mathcal{S}_k^*$. Now starting from that state $\mathbf{s}_0$, let the trajectory of states taken by $\pi$ be $\{\mathbf{s}_0, \mathbf{s}_1, ..., \mathbf{s}_T = g\}$. Note that since $d^{\pi}(\mathbf{s}_0, \mathbf{g}) > d^*(\mathbf{s}_0, \mathbf{g}) = K$, $T > K$. Let $\hat{\pi}$ be the policy such that it agrees with $\pi^*$ on $\mathbf{s}_0$ and agrees with $\pi$ everywhere else. At the first step, $\hat{\pi}$ lands on state $\mathbf{s}'_1$. Since $\mathbf{s}_0$ is $K$ steps away from $\mathbf{g}$ under $d^*$, $\mathbf{s}'_1$ must be $K - 1$ steps away under $d^*$ and $\mathbf{s}'_1 \in \mathcal{S}_{K-1}^*$. Therefore, since $\pi$ and $\pi^*$ agrees on all states that are less than $K$ steps away from goal

---

[4]Of course, the actual DDL algorithm interleaves policy updates and distance updates. In this appendix, we analyze the "policy iteration" variant that optimizes the policy to convergence, but the result can likely be extended to interleaved updates in the same way that policy iteration can be extended into an interleaved actor-critic method.

$\mathbf{g}$, $\hat{\pi}$ would take the same action as $\pi^*$ and hence take another $K - 1$ steps to goal $\mathbf{g}$. Now let us denote the trajectory taken by $\hat{\pi}$ as $\{\mathbf{s}'_0 = \mathbf{s}_0, \mathbf{s}'_1, ..., \mathbf{s}'_K = g\}$. We compare the discounted sum of rewards of $\pi$ and $\hat{\pi}$ under the reward function $r_\mathbf{g}(\mathbf{s}, \mathbf{a}) = -d^\pi(\mathbf{s}, \mathbf{g})$.

$$
\begin{aligned}
\mathcal{L}_\pi(\pi) = \sum_{i=0}^{T-1} \gamma^i d^\pi(\mathbf{s}_i, \mathbf{g}) &= d^\pi(\mathbf{s}_0, \mathbf{g}) + \sum_{i=1}^{T-1} \gamma^i d^\pi(\mathbf{s}_i, \mathbf{g}) \\
&= d^\pi(\mathbf{s}_0, \mathbf{g}) + \sum_{i=1}^{T-1} \gamma^i (T - i) \geq d^\pi(\mathbf{s}_0, \mathbf{g}) + \sum_{i=1}^{K-1} \gamma^i (K - i) \\
&= d^\pi(\mathbf{s}_0, \mathbf{g}) + \sum_{i=1}^{K-1} \gamma^i d^\pi(\mathbf{s}'_i, \mathbf{g}) = \sum_{i=0}^{K-1} \gamma^i d^\pi(\mathbf{s}'_i, \mathbf{g}) = \mathcal{L}_\pi(\hat{\pi})
\end{aligned}
\tag{10}
$$

Therefore, we can see that $\hat{\pi}$ is a better policy than $\pi$. Then the optimal policy $\pi'$ under this reward must be different from $\pi$ on at least one state. Hence $d^\pi(\mathbf{s}, \mathbf{g}) \neq d^{\pi'}(\mathbf{s}, \mathbf{g})$.

We've now reached the conclusion that if $d^{\pi'}(\mathbf{s}, \mathbf{g}) \neq d^*(\mathbf{s}, \mathbf{g})$, then $d^{\pi'}(\mathbf{s}, \mathbf{g}) \neq d^\pi(\mathbf{s}, \mathbf{g})$. Hence, by contraposition, if $d^{\pi'}(\mathbf{s}, \mathbf{g}) = d^\pi(\mathbf{s}, \mathbf{g})$, then it must be that $d^{\pi'}(\mathbf{s}, \mathbf{g}) = d^*(\mathbf{s}, \mathbf{g})$. Our proof is thus complete.

□

## C  DIDACTIC EXAMPLE

Our didactic example involves a simple 2D point robot navigating an S-shaped maze. The state space is two-dimensional, and the action is a two-dimensional velocity vector. This experiment is visualized in Figure 7. The black rectangles correspond to walls, and the goal is depicted with a blue star. The learned distance from all points in the maze to the goal is illustrated with a heat map, in which lighter colors correspond to closer states and darker colors to distant states. During the training, the initial state is chosen uniformly at random, and the policy is trained to reach the goal state. From the visualization, it is apparent that DDL learns an accurate estimate of the true dynamical distances in this domain. Note that, in contrast to naïve metrics, such as Euclidean distance, the dynamical distances conform to the walls and provide an accurate estimate of reachability, making them ideally suited for reward shaping.

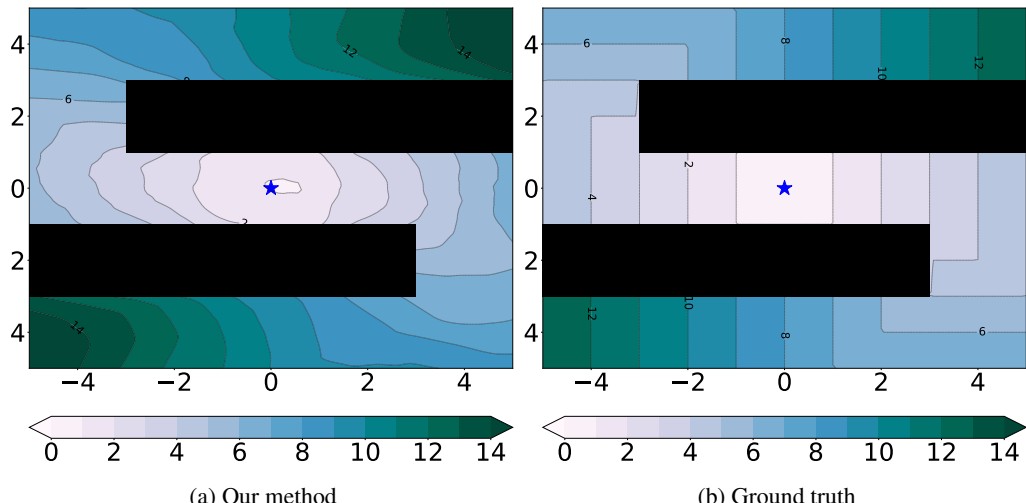

(a) Our method        (b) Ground truth

Figure 7: Evaluation of the learned distance in a 2D point environment. The state is the xy-coordinates of the point, and action corresponds to 2D velocities. The black bars denote walls, blue star is a goal state, and the heat map denotes the estimated distance to the goal. (a) Our method learns an accurate estimate of the shape of the distance function. (b) Ground-truth distance.

## D    PREFERENCE QUERIES FOR REAL-WORLD DCLAW EXPERIMENT

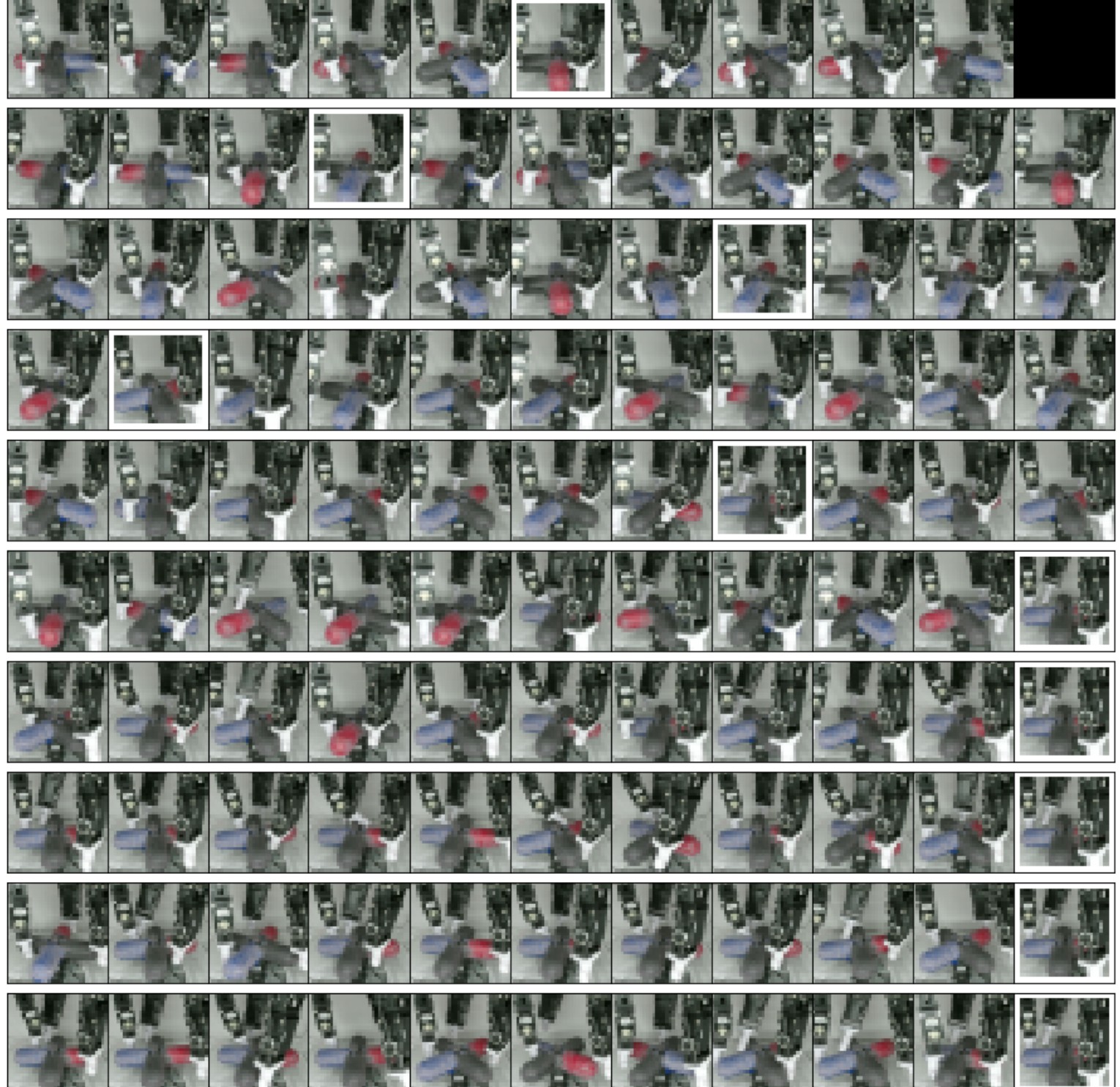

Figure 8: Human preference queries for the vision-based DClaw experiment presented in Section 6.1. Each image row presents the set of images shown to the human operator on a single query round. On each row, the first 10 images correspond to the last states of the most recent rollouts and the right-most image corresponds to the last goal. For each query, the human operator picks a new goal by inputting its index (between 0-10) into a text-based interface. The goals selected by human are highlighted with white borders.

# E   TECHNICAL DETAILS

All our experiments use Soft Actor-Critic as the policy optimizer, trained the default parameters by provided by the authors in (Haarnoja et al., 2018c).

For all of the tasks, we parameterize our distance function as a neural network. For state-based tasks, we use feed-forward neural networks with two 256-unit hidden layers. For the vision-based tasks we add a convolutional preprocessing network before these fully-connected layers, consisting of four convolutional layers, each with 64 3x3 filters. Both cases use Adam optimizer with learning rate 3e-4 and TensorFlow's default momentum parameters. The image observation for all the vision-based tasks are 3072 dimensional (32x32 RGB images).

Most important hyperparameters that we swept over in the final experiments, namely the size of the on-policy pool for training the distance function and the number of gradient steps per environment samples, are presented in Table 1 below:

| Environment | gradient steps per environment steps | on-policy pool size |
|---|---|---|
| InvertedDoublePendulum-v2 | $1/64$ | $100k$ |
| Hopper-v3 | $1/64$ | $16k$ |
| HalfCheetah-v3 | $1/16$ | $16k$ |
| Ant-v3 | $1/64$ | $10k$ |
| DClaw (both state and vision) | $1/16$ | $100k$ |

Table 1: Distance estimator hyperparameters.

For the DDLUS goal proposals, we consider all the samples in the distance on-policy pool as the goal candidates. For DDLfP, we present the operator the last states ($s_{T-1}$) of the last $N$ episodes, where $N = 5$ for all the simulated experiments, and $N = 10$ for the hardware DClaw.

As discussed in Section 5, for both DDLUS and DDLfP, the agent needs to explore in the vicinity of the goal state. In practice, we implement this by switching to a random uniform policy after 0.9T timesteps of each episode, where T is the maximum episode length (1000 for all the mujoco tasks and 200 for the DClaw task).

