# OpenReview forum: "Dynamical Distance Learning for Semi-Supervised and Unsupervised Skill Discovery"
_ICLR.cc/2020/Conference — Accept (Poster)_

### Official Review · AnonReviewer1 · 2019-10-14
**Official Blind Review #1**

**Rating:** 6

**Review:**

This paper presents an approach to do reinforcement learning without reward engineering. Instead, steps to reach goals are used to update policy.

Here are some questions:
(1) The reason to use cumulative distance instead of greedily chooses the shortest distance is to avoid risky state. But it is also mentioned in the paper that the experiments in the paper all have deterministic dynamics (and one of the assumptions in Appendix B). Can the authors elaborate on this? For example, how is the performance of a policy that just directly minimizes the distance function? Instead of using the made-up toy example.

(2) For the locomotion task examples, I am confused as to why choosing the state with the highest immediate reward will be a good preference state. For example, a humanoid that falls forward can have high forward velocity while has no chance for recovery. While the performance seems reasonable, the systems presented are mostly stable systems (with the exception of hopper). It will be nice to show similar performance in more unstable systems, e.g, walker2d, humanoid.

(3) Some assumption in the proof in appendix B is not true. d(g, g) is not 0 for all dynamical systems. For example, a hopper with a state that has none zero forward velocity will take more than 0 steps to return to the same state.

(4) “Because the trajectories have a finite length, we are effectively ignoring the cases where reaching sj from si would take more than T − i steps, biasing this estimate toward zero, but since the bias becomes smaller for shorter distances.” I am confused about this sentence, what does it mean? For example, if the dynamical system is the hopper, (s_i, s_{i+1}) will make d(s_i, s_{I+1}) be 1, and for all we know d(s_{i+1}, s_i) can be huge. How does this sentence apply in this situation?

(5) An appendix with hyperparamters will be helpful. For example, how much data are needed for updating the distance function?

(6) “We also evaluated DDLUS on the InvertedDoublePendulumv2 domain, where the task is to balance a pole on a cart. As can be seen from Figure 6, DDLUS can efficiently solve the task without the true reward, as reaching dynamically far states amounts to avoiding failure as far as possible.” Why reaching dynamically far states is equivalent to avoiding failure? Shouldn’t the policy learn to stay as close to the initial state as possible?

Despite the questions raised above, the techniques are interesting, simple, and effective, as demonstrated on some image-based tasks on real robots. While I understand that this method will be very effective for relatively static tasks like the valve turning, I fail to see how it can apply in unstable locomotion tasks, as mentioned in some of the questions above.

Normally I will give rating 5 to this paper, but the system doesn't give me this option, so I have to lower it to 3 instead.

**Experience Assessment:**

I have published one or two papers in this area.

**Review Assessment: Checking Correctness Of Derivations And Theory:**

I carefully checked the derivations and theory.

**Review Assessment: Checking Correctness Of Experiments:**

I assessed the sensibility of the experiments.

**Review Assessment: Thoroughness In Paper Reading:**

I read the paper at least twice and used my best judgement in assessing the paper.

---

> ### Author Response · Authors · 2019-11-08
> **Author Reply for Official Blind Review #1**
>
> Thanks a lot for the detailed and insightful comments! Hopefully the answers below provide some clarification. Please let us know if any further clarification is needed or other questions arise.
>
> (1) Thanks for pointing out the lack of experiments with stochastic dynamics. It is true that our reasoning with the three-state stochastic example does not justify the use of cumulative distances in the simulated experiments and we don't fully understand why cumulative distances work better even in those cases. We now explicitly mention this at the end of Section 4.3. Note that Hindsight Experience Replay method, shown in Section 6.2., greedily minimizes the distance function and performs much worse compared to DDL, even in tasks with deterministic dynamics. We observed the same behavior across all the other tasks as well. We also experimented with the setup you mentioned, where the distance is learned with a supervised loss and then greedily minimized, but never managed to get the setup working reliably. We will add this case as one of the ablations in Section 6.2 shortly and comment below when we have updated the paper with them.
>
> (2) Our description for the DDLfP goal selection for the locomotion tasks is definitely confusing, and we have clarified this in the Section 6.2. To summarize: “[preferences] are provided synthetically, choosing the state that has progressed the largest distance from the initial state in the desired direction, i.e. the state with largest x-coordinate value.”
>
> (3) The proof in Appendix B assumes that g is a terminal state. Therefore, the expectation in our definition of the distance (Equation 2) is taken over trajectories that start in g and terminate immediately, and thus the sum reduces to zero. We now mention this assumption in Appendix B in the paper.
>
> (4) By this we mean that even though the expectation in Equation 2 is taken over infinitely long trajectories (unless they terminate by reaching a terminal state), in practice, we have a finite horizon and truncate the sum at the end of it, and we thus ignore the terms far in the future (H < t). These are all positive terms, creating a negative bias to our distance estimator.
>
> (5) Not adding the hyperparameters and other technical details was an oversight from our end. We have included all the details in Appendix E in the latest version of the paper and will also release the code upon acceptance of the paper.
>
> (6) DDLUS chooses the goal to be the state that is farthest away from the initial position. Now, because in InvertedDoublePendulum-v2, the episodes start with the pole upright and are terminated whenever the pole falls over to a certain angle, then the states that lead to premature termination will be closer (have distance less than the horizon length) than those states that keep the pole balanced for the maximum horizon. Thus the policy learns to balance the pole, but the cart’s x-position might still be different from the initial position.
>
> We're happy to provide more information if needed!

---

> > ### Comment · AnonReviewer1 · 2019-11-11
> > **More question**
> >
> > Here are some follow-up questions:
> > (2) Here you choose the state with largest x-coordinate value, does this mean x is an input to the policy? If so, this input is unbounded and if you keep the policy running it will eventually go to some value the policy cannot handle. And again, a state that falls forward at high velocity can have the largest x-coordinate value and is a really bad goal state. And from the video, it seems to agree with my assumption that this method doesn't work for unstable locomotion task, only one of the four hoppers finally hops.
> > (3)I understand that you want to terminate once the goal is reached for tasks like the valve turning. But I cannot say the same for the locomotion tasks, where a policy should just keep going forever. So I'm not sure this is a generally applicable assumption. Furthermore, in the paper you mention "and only a single query for the InvertedDoublePendulum-v2, as the initial state and the goal states coincide". By your assumption, the trajectory length is 0 for all roll-outs.
> > (6) The same confusion remains, also mentioned in my previous point. I don't understand why not just learn a policy that will just remain in the initial position.
> >
> > Non-technical issue: the video of the half cheetah is really hard to watch because of the high-speed changes of the background.
> >
> > Again, I totally understand how this will work for the valve turning tasks, but I don't see how it can apply to the standard locomotion benchmarks in mujoco, unless further evidence suggests otherwise.

---

> > > ### Author Response · Authors · 2019-11-13
> > > **Author Reply for Official Blind Review #1**
> > >
> > > Thanks for the follow-up questions!
> > >
> > > (2) Regarding the unbounded x-coordinate and the problem of goal-conditioned locomotion: You’re correct that the current x-coordinate is an input to the policy. It is also true that the locomotion tasks/goals like "run as far as possible" don't fit into the goal-conditioned framework directly and can be a little confusing. We have revised Section 6.2 to discuss this limitation. To summarize: One way to think about it is that, in this particular case of locomotion tasks, we could fix the issue by considering a case where the “ultimate” task is to reach a specific goal, i.e. the operator would always choose the goal to be the state that is closest to the “ultimate task goal”. In that case, we can see the locomotion task to be the limit case where the “ultimate goal” is as far as possibly reachable within the maximum episode length. Our motivation to evaluate the method in the locomotion tasks was to provide a direct comparison to prior works such as DIAYN and Christiano et al.
> > >
> > > > From the video, it seems that only one of the four hoppers finally hops
> > >
> > > Our apologies, the website was not updated to include the latest versions of our policies and thus performance in those videos was a bit out-of-date. We’ve updated the videos for Hopper-v3 and HalfCheetah-v3 to match our latest results. To summarize: for DDLUS Hopper-v3, 7/10 seeds actually learn to jump at least 3 reasonable jumps (either forward or backward), none of them showing the behavior of falling into a “bad” local minima; for DDLfP Hopper-v3, 5/5 policies manage to jump forward for pretty much the whole episode. Videos for other seeds will be available here: https://drive.google.com/open?id=1s7nvRzjd9vdlxPl6tZ0myT2MkZ5VqvXy.
> > >
> > > > a state that falls forward at high velocity can have the largest x-coordinate value and is a really bad goal state.
> > >
> > > Even when the state with the highest x-value at some point happens to be a “bad” state, the stochastic policy will explore around the current goal and thus reach other states that result in higher “long-term” x-values. As discussed in Section 5 and Appendix E, we additionally allow the policy to take uniformly random actions at the end of each episode for even better exploration. Our case is analogous to the traditional RL framework: in the case of Hopper-v3, for example, the policy might find a state that has a huge velocity in both the *positive* x-direction and the *negative* y-direction (i.e. falling down while moving forward). This state would yield a high immediate reward (because of the high x-velocity) right before the episode terminates (due to falling down). The way the RL agent still learns to avoid these local minima is by exploring states that results in a higher long-term reward.
> > >
> > > (3, 6) You’re correct that the semi-supervised pendulum case is a bit silly from the theoretical perspective (since it would indeed terminate immediately). These experiments don’t fully follow the theory in the sense that the goal $g$ is not necessarily a terminal state.
> > >
> > > > The video of the half cheetah is really hard to watch
> > > We’ve re-uploaded the HalfCheetah videos with a bit nicer camera angle and slower frame rate.
> > >
> > >
> > > We’re happy to elaborate further so keep the questions coming!

---

> > > > ### Comment · AnonReviewer1 · 2019-11-14
> > > > **thanks for the clarification**
> > > >
> > > > Thanks for the clarification. I have updated my score.

---

### Official Review · AnonReviewer2 · 2019-10-23
**Official Blind Review #2**

**Rating:** 6

**Review:**

Summary
The authors propose learning the expected # of time steps to reach a given goal state from any other state which they call: dynamical distances. The motivation suggests that such an ability would help the agent in reaching new goal states and therefore in learning complex tasks. Task goal is given with a small amount of preference supervision. The claim is that dynamical distances can be used in a semi-supervised regime, where unsupervised interaction with the env is used to learn the dynamical distances. Evaluation is done on a real-world robot and in simulation, the method can learn to turn a valve with a real-world 9-DoF hand, using raw image observations and just ten preference labels, without any other supervision. The ideas presented in this work are interesting, but I have some major concerns (please see detailed comments below). The paper is well written and is mostly easy to understand.

Detailed comments:
The learning process follows two steps:
-distance evaluation step: learn a policy specific dynamical distance parameterized by \psi: by computing the expected # of time steps it took for \pi to reach the goal state.
-policy improvement step: use the learned distance fn to optimize a policy to reach the goal

One weakness of the proposed approach is that the supervised regression step requires an on-policy experience which can be very expensive as pointed out by the authors as well. The authors suggest that because they use this as an intermediate representation this does not impact learning. Wouldn’t it be possible to *simultaneously* learn the dynamical distance during the off-policy policy-learning step?

The second weakness of the proposed approach is that in the policy improvement step it is assumed that a goal state is given. It is also weird to use goals that are already reachable. Please provide clarifications.

Algorithm 1 learns the distance corresponding to the current policy. Wouldn’t it be a more generalized approach to learn the dynamical distance irrespective of the policy i.e., not policy-specific? Or even better, would it be possible to instead learn the dd applicable to a class of policies? Assuming behaviourally similar policies can have the same dynamical distance?

On an intuitive level, it would be useful for the reader to make concrete how is this approach different from learning goal conditioned policies? The dynamical distances reflect the distance to the goals: goals are either 1) sampled from an experience replay buffer, 2) given by user preference, 3) chosen more smartly by the dynamical distance learned.

It is not clear if in sec 4.3 “the optimal dynamical distance will be 2”, but since the Eq.2. use the expected distance, does the above statement hold?

Empirical analysis: The authors aim to address three questions through the experiments: Regarding Q2: Is DDL applicable to real-world, vision-based robotic control tasks? Fig 1 shows interesting tasks and a useful contribution.

In terms of learning from human preferences; the experiments do not seem sufficient and need more comparative analysis. In particular, the authors should compare to other well-known techniques where learning from human preferences is performed such as [1], [2].

The same is the case with unsupervised skill acquisition, the authors only compared to Diayn. I would suggest comparing to other relevant baselines such as [3], [4], [5].  Fig 6 primarily suggests the obvious that Diayn maximizes for diversity while DDL maximizes for distance. Moreover, even this comparison seems incomplete as I am not sure why the top row in Fig 6 only shows the proposed approach and not the baseline (diayn). Unsupervised skill acquisition makes it even more important to understand what is the nature of skills learned. I find that analysis is missing here as well, and would add completeness to these results.

Considering the paper’s claims are heavily based on the ability to learn skills, there is no qualitative analysis of what kind of skills are learned. For e.g. it is mentioned that DDL method that can learn a 9-DoF real-world dexterous manipulation task, but what are the exact skills learned here per task? It would be useful to do a more rigorous analysis of each skill learnt. “Videos of the learned skills can be found on the project website: https://sites.google.com/view/skills-via-distance-learning.” However, the link throws a 404 error.

Overall:
The paper is well written, real-world experiments are a ++, interesting mix of ideas along with applicability to multiple problems. The main experiment is shown with preferences where comparisons do not seem enough and the paper could be made stronger by thorough comparisons.

[1] Christiano, Paul F., et al. "Deep reinforcement learning from human preferences." Advances in Neural Information Processing Systems. 2017.
[2] Ashesh Jain, Brian Wojcik, Thorsten Joachims, and shutosh Saxena. Learning trajectory preferences for manipulators via iterative improvement. In NIPS, 2013.
[3] Gupta, Abhishek, et al. "Unsupervised meta-learning for reinforcement learning." arXiv preprint arXiv:1806.04640 (2018).
[4] Achiam, Joshua, et al. "Variational option discovery algorithms." arXiv preprint arXiv:1807.10299 (2018).
[5] Karol Gregor, Danilo Rezende, and Daan Wierstra. Variational Intrinsic Control. pages 1–15, 2016.


**Experience Assessment:**

I have published one or two papers in this area.

**Review Assessment: Checking Correctness Of Derivations And Theory:**

I carefully checked the derivations and theory.

**Review Assessment: Checking Correctness Of Experiments:**

I carefully checked the experiments.

**Review Assessment: Thoroughness In Paper Reading:**

I read the paper thoroughly.

---

> ### Author Response · Authors · 2019-11-08
> **Author Reply for Official Blind Review #2**
>
> Thanks a lot for the insightful comments! Hopefully the answers below address all the feedback and questions. Please let us know if any further clarification is needed or other questions arise.
>
> > Wouldn’t it be possible to *simultaneously* learn the dynamical distance during the off-policy policy-learning step?
>
> This is actually exactly what the “DDLfP w/ TD update” and “Hindsight Experience Replay” methods in the Section 6.2. ablations do. Unfortunately, such off-policy distance updates are substantially harder from an optimization perspective. As our results in Figure 5 show, the dynamical distances (trained on-policy) allow the policy to reach the goal when learning both from state and from images, whereas both of the comparisons struggle when running on high-dimensional images. The off-policy version (with TD update) fails even on the low-dimensional state space.
>
> > The second weakness of the proposed approach is that in the policy improvement step it is assumed that a goal state is given.
>
> Indeed the policy improvement step requires that the goal is given. Alternatively, one could condition the policy on the goal and learn to minimize the distance to all possible states. In our experiments, however, we found learning goal-conditioned policies substantially harder, and therefore present a method that optimizes a policy to reach a single given goal instead. As a future work, we plan to lift this constraint by learning a state-conditioned policy that can reach a set of interesting goals rather than a single goal.
>
> > It is also weird to use goals that are already reachable.
>
> Choosing a goal that the policy can already reach might indeed at first appear strange, but it turns out to yield efficient directed exploration. If we first use the policy to reach a known goal as quickly as possible — in contrast to reaching the goal by random exploration — we can spend much more time exploring the vicinity of that goal. We clarified this in the beginning of Section 5 in the paper.
>
> > Wouldn’t it be a more generalized approach to learn the dynamical distance irrespective of the policy i.e., not policy-specific? How is this approach different from learning goal conditioned policies?
>
> We are interested in learning a policy that can reach a given goal as quickly as possible. Simply learning the shortest distance between any two states (as in [6]) does not immediately yield an optimal policy for reaching any states, but would also requires conditioning the policy on the goal state (which we don't do). As mentioned above, we found learning goal-conditioned policies hard, and also unnecessary for the tasks that we considered in this work (e.g. rotating a valve to a specific orientation).
>
> > It is not clear if in sec 4.3 “the optimal dynamical distance will be 2”, but since the Eq.2. use the expected distance, does the above statement hold?
>
> Consider a policy that always chooses $s_4$ from the initial state $s_0$. Then the distance from $s_0$ to $g$ for this particular policy is 2, because the expectation in Equation 2 only considers the successful trajectories that avoid the terminal state ($s_T$). On the other hand, a policy that always chooses $s_1$ from the initial state, takes 3 steps to reach the goal, and is thus not optimal. The optimal distance (the distance under optimal policy) is thus 2.
>
> > In terms of learning from human preferences. The authors should compare to other well-known techniques where learning from human preferences is performed such as [1], [2].
>
> In Section 6.2 and Figure 4, we already have a direct comparison with Christiano et al. [1] on the locomotion tasks.
>
> > The same is the case with unsupervised skill acquisition, the authors only compared to Diayn. I would suggest comparing to other relevant baselines such as [3], [4], [5].
>
> [3] actually uses vanilla DIAYN for the unsupervised task acquisition. We’ll do our best to add a comparison with another baseline before the review period ends.
>
> > The top row in Fig 6 only shows the proposed approach and not the baseline (diayn).
>
> We obtained the DIAYN performance numbers for Figure 6 directly from the authors, who did not have a learning curve that they could provide. We will rerun the DIAYN ourselves and add these in the plots.
>
> > It is mentioned that DDL method that can learn a 9-DoF real-world dexterous manipulation task, but what are the exact skills learned here per task? The link throws a 404 error.
>
> The link error is unfortunately caused by the way openreview website parses links and should work if the dot at the end of the url is omitted (https://sites.google.com/view/skills-via-distance-learning). See the skills learned both for the locomotion tasks and the hardware DClaw in the videos available at the website.
>
> [6] Kaelbling, Leslie Pack. "Learning to achieve goals." IJCAI. 1993.

---

> > ### Comment · AnonReviewer2 · 2019-11-14
> > **Follow-up comments**
> >
> > Thank you for the clarifications.
> >
> > -Re "As mentioned above, we found learning goal-conditioned policies hard, and also unnecessary for the tasks that we considered in this work (e.g. rotating a valve to a specific orientation)."
> > I am not so sure I fully follow the latter part. One could argue that reaching a specific orientation of the valve is for instance a perfectly well defined goal.  Could you please give some more clarity on how is this different from learning policies which reach sampled/given goals such as FuN (Feudal networks for instance). The policy improvement step requires that the goal is given as well.
> >
> > -Re unsupervised skill acquisition, the authors only compared to Diayn: Would you be able to comment on the performance from other baselines apart from Diayn?
> >
> > -Re The top row in Fig 6, It would be nice to see the baseline results as well. I strongly recommend these additional experiments to strengthen the empirical results and overall value to this work.
> >
> > -Re the qualitative videos: I appreciate random seeds and not cherry picking. I am not so sure what to read in the Hopper videos: I see some times the hopper hops but not robust across seeds. Similarly in antv3, the skills do not appear to be robust across different ants shown. Is there a particular reason you suspect the cause towards the instability across seeds?

---

> > > ### Author Response · Authors · 2019-11-15
> > > **Author Reply for Official Blind Review #2**
> > >
> > > Thanks a lot for the additional comments!
> > >
> > > > Could you please give some more clarity on how is this different from learning policies which reach sampled/given goals such as FuN.
> > >
> > > You are right in that in our case, the task the policy is trying to solve correspond to reaching a specific goal state (that is evolving throughout the training) as quickly as possible. However, the policy is not goal-conditioned (the goal is not an explicit input to the policy) but instead, it learns to reach one goal at a time. However, the DDL framework can readily be turned into goal-conditioned by conditioning the policy and the distance function by the goal. That is, currently our policy is of the form $pi(a|s)$ and it can only reach one goal at a time. In the goal-conditioned case, we would condition the policy with a goal $g$ by providing it as an input to it, as $pi(a|s, g)$, meaning that the same policy can learn to reach many goals depending on which goal we condition it on. Note, that once we change our policy to be goal-conditioned, the distance between two states ($s1$, $s2$) also becomes dependent on the goal $g$ the policy was trying to reach, which is why we would also need to condition the distance function with $g$, i.e. $d(s1, s2 | g)$. In feudal networks, the lower-level policy is of the latter, goal-conditioned form, i.e. it can learn to reach different goals received from the higher-level policy.
> > >
> > > Training goal-conditioned policies is somewhat orthogonal (albeit definitely related) to our work, and has its own challenges, which is why we decided to use regular policies for DDL. But as mentioned, in theory, DDL can be also used for training goal-conditioned policies.
> > >
> > > > Would you be able to comment on the performance from other baselines apart from Diayn?
> > >
> > > To our knowledge, not many methods do unsupervised skill discovery, but it looks like the ones that were mentioned in the previous comment could be potential candidates for baselines. We’re currently running an additional comparison with VALOR [1]. Unfortunately, it’s unlikely that we’ll have these ready before the rebuttal closes (the runs typically take about two days to finish for Ant and HalfCheetah), but we’ll add them in the final.
> > >
> > > Thanks for pointing this out!
> > >
> > > > Re The top row in Fig 6, It would be nice to see the baseline results as well.
> > >
> > > We’re currently rerunning the DIAYN experiments and will add the learning curves once the runs are finished. For now, we’ve extended Figure 6 to show the final performance of the DIAYN policies extracted from the data received from the DIAYN authors.
> > >
> > > > I am not so sure what to read in the Hopper videos: I see some times the hopper hops but not robust across seeds.
> > >
> > > For the unsupervised DDLUS, The behavioral difference across the seeds is not exactly due to instability, but more about the nature of the unsupervised skill discovery. Since the policy does not receive a task reward, it will learn a different skill on each training run -- sometimes the policy learns to hop and in some other cases it might do something very different. For DDLUS, 7/10 seeds hop (some backwards and some forwards), whereas 3 of the seeds have learned other behaviors. On the other hand, for the supervised DDLfP, 5/5 hopper seeds consistently hop forward, which is due to the supervision it receives through the goals.
> > >
> > > Note that all the seeds in the summary videos on the website are randomly picked, and videos for the rest of the seeds are also available in this link: https://drive.google.com/open?id=1s7nvRzjd9vdlxPl6tZ0myT2MkZ5VqvXy.
> > >
> > >
> > > [1] Achiam, Joshua, et al. "Variational option discovery algorithms." arXiv preprint arXiv:1807.10299 (2018).

---

### Official Review · AnonReviewer3 · 2019-10-24
**Official Blind Review #3**

**Rating:** 6

**Review:**

I'm afraid I found this paper somewhat confusing and hard to see the big picture but I also acknowledge that I am not an expert in deep, model-free RL and my RL experience is mostly in model-based RL and I am happy for this to be taken into consideration when evaluating my review - apologies if I miss something that is well known in that community.


After a few readings my understanding of the paper is that there is a desire to develop a proxy (or basis) that is first determined in a reward free setting, agnostic to a goal. Subsequently, this proxy can then be combined with a goal and improve efficiency in developing a suitable policy for that goal rather than starting from scratch with something like Q-learning. Is this the case or have I misunderstood something?

If this is the case, I can envisage an argument for such a setting although it surprises me that something similar has not been performed previously - the approach seems to be to be quite similar to a model-based setting where this distance regressor is in many ways similar to a model in that it is trained from unsupervised trajectories and effectively encodes the transitions between states? Given that the distance in (3) is essentially trained from trajectories with a fixed reward structure in (2) I'm very surprised that a Q-Learning approach doesn't offer similar performance when the task is specified - can the authors please provide more details as to why this approach is expected to be so superior to Q-learning?

The distances between states from the regressor is policy dependent and then this distance is used to inform the policy update - how can we be certain that this alternating optimization will not fall into a local minimum depending on the initial policy? None of the experiments seem to check for this?

The experiments are quite specific, both in terms to the particular experiment and very specific previous work, which makes it hard to judge empirically the merits of the approach. Again, I would defer to others with more experience in this field to know whether or not this is standard practice?


- Something that I think needs to be changed is the continual use of the term distance when it is not a valid mathematical distance (this point is noted by the authors but then the term is used continuously and the notation used would be standard notation for an actual distance). Would it not be more appropriate to call it a dissimilarity?

Other notes:

- In section 4.4 I think it is really bad practice to suggest that a fixed number of stochastic gradient descent steps avoid over-fitting - I know of nothing that guarantees this statement.

- I think it is not a given that it is easier to look through trajectories and express preferences over certain video frames as opposed to taking images of the desired goal at the start - are the experiments comparing like with like? I think both methods could be presented with the same supervision to make the comparisons fair.

- I find the term skill discovery strange - would it not be more helpful to include model-free RL in the title?

**Experience Assessment:**

I do not know much about this area.

**Review Assessment: Checking Correctness Of Derivations And Theory:**

I assessed the sensibility of the derivations and theory.

**Review Assessment: Checking Correctness Of Experiments:**

I assessed the sensibility of the experiments.

**Review Assessment: Thoroughness In Paper Reading:**

I read the paper at least twice and used my best judgement in assessing the paper.

---

> ### Author Response · Authors · 2019-11-08
> **Author Reply for Official Blind Review #3**
>
> Thanks a lot for the constructive feedback and for taking the time to review our paper! Hopefully our answers below clarify the topic a bit. Please let us know if any further clarification is needed or other questions arise.
>
> > There is a desire to develop a proxy (or basis) that is first determined in a reward free setting, agnostic to a goal...
>
> We are indeed operating in a completely reward-free setting. To help better understand the situation, imagine that you were given a distance function $d(s1, s2)$ that encodes the number of environment steps (i.e. number of actions) it takes to reach state $s2$ from $s1$. In this case, you could train a goal-conditioned policy $\pi(a|s,g)$ that minimizes the number of steps to get from any state $s$ to goal $g$, by just considering the distance to the goal as the negative reward (i.e. $r(s,a) = -d(s,g)$).
>
> Now, the obvious problem is that we don't have such a distance function in the general case. In this work, we propose a way of learning this distance function with a simple supervised learning method. We then propose two ways (DDLUS and DDLfP) to choose the goals $g$.
>
> > It surprises me that something similar has not been performed previously…
>
> As we discuss in Section 2, similar techniques have been proposed previously. Many of these works learn the distance function by using temporal difference updates or learn a value function that implicitly captures the distance under a user-specified low-dimensional goal representation. Unfortunately, such methods are substantially harder from an optimization perspective. As our results in Figure 5 show, the dynamical distances (trained on-policy) allow the policy to reach the goal when learning both from state and from images, whereas both of the comparisons struggle when running on high-dimensional images. The off-policy version (with TD update) fails even on the low-dimensional state space.
>
> > How can we be certain that this alternating optimization will not fall into a local minimum depending on the initial policy?
>
> As all our experiments are carried out using neural network approximators, there unfortunately aren’t guarantees for avoiding local minima. This is an issue not specific to our work, but is also present in most of the other deep reinforcement learning methods.
>
> > The experiments are quite specific.
>
> Based on the two other reviews, a couple of additional experiments would strengthen the empirical results: A better comparison with another unsupervised method and also an ablation where we train the policy by greedily minimizing the distance function learned in supervised fashion. Right now in Section 6.2. we only show results for case where greedily minimize the distance function that is trained with TD-update. We will update the paper with these experiments and notify you when it’s done.
>
> > Something that I think needs to be changed is the continual use of the term distance when it is not a valid mathematical distance. Would it not be more appropriate to call it a dissimilarity?
>
> We decided to use the term distance because it has already been widely adopted for this purpose in the literature such as [1]. Dissimilarity would suggest how (dis-)similar two environment states are, whereas we try to learn a function that encodes how (temporally) far the two states are apart, which is why we believe the term distance is better suited for this case.
>
> > In section 4.4 I think it is really bad practice to suggest that a fixed number of stochastic gradient descent steps avoid over-fitting - I know of nothing that guarantees this statement.
>
> Thanks for pointing this out. We agree that there are no guarantees for this. In practice, we consider the number of gradient steps as a hyperparameter. We have clarified Section 4.2. and added Appendix E for the details of the hyperparameter.
>
> > I think it is not a given that it is easier to look through trajectories and express preferences over certain video frames as opposed to taking images of the desired goal at the start.
>
> We agree that the difficulty depends on the task in hand, and thus the statement might not hold in the general case, but preferences require a user to simply select one state out of a small set, which can be done with a button press and done e.g. remotely, while providing a goal example requires physically arranging a scene and taking a picture. Therefore, we believe it is reasonable to claim that in many cases, preferences are less labor-intensive, though this is of course not universal.
>
> > I find the term skill discovery strange - would it not be more helpful to include model-free RL in the title?
>
> We believe that it’s not unreasonable to call the acquired behaviors skills. DDL isn’t limited to model-free RL, and we could choose to optimize our policy in any on-policy RL method as well. We chose to use Soft Actor-Critic due to its good sample efficiency and ease of tuning.
>
>
> [1] Kaelbling, Leslie Pack. "Learning to achieve goals." IJCAI. 1993.

---

> > ### Comment · AnonReviewer3 · 2019-11-15
> > **Updating scores**
> >
> > Thank you for taking the time to provide the detailed comments and taking suggestions on board. I think I see the motivation more clearly now and some of the convention confusions are probably related to differing fields. I will update my score and am quite happy to go with the consensus of the other reviewers (who are more on topic as well).

---

### Public Comment · ~Academic_Anony_Mous1 · 2019-12-26
**Clarifications in definition and Appendix A**

Hi,

Can you please clarify whether the distance as defined in equation (2) computes the average number of steps to reach a state s_j from s_i for the first time or all visits? For example, in the sample trajectory s_1, s_2, s_3, s_4, s_5, s_3, s_6 is the distance between s_1 and s_3 2? or is it the average of all of {2, 3, 5}?

In the policy improvement section you provide an example of a pathological example and mention that being greedy with the distance function (by considering the negative value function) leads to undesired behaviour. It it is mentioned in the same section that minimizing the cumulative distances solves the pathological behaviour. The proof for which is provided in Appendix A.

The proof presented in Appendix A assumes that all episodes reach the goal the state and the goal state is the terminal state. To rephrase it, the goal state is the only absorbing state. This assumption is not satisfied in the pathological example since s_T is also an absorbing state in addition to the goal state g. So, the connection between the proof in Appendix A and the pathological example is unclear to me. Can you please explain?

My understanding of why the pathological behaviour occurs is because estimating the distance using the proposed approach does not correspond to estimating the value function given by the binary reward function which gives reward 1 for non goal states and 0 reward for the goal state. It is not a value function because, as stated in the paper, only the states occurring in the goal reaching trajectories are used to compute the distance whereas a value function uses all the trajectories possible from any given state. For example, the "value" of s_4 given by DDL is 1 whereas the true value of s_4 is infinity because S_T is absorbing.

Since DDL doesn't capture the distance using unseen trajectories, the cumulative distance function also doesn't capture the distance from unseen trajectories. Hence, minimizing the cumulative distance suffers from the same problem and therefore the pathological behaviour is not overcome.

For example, considering the MDP provided in page (4), d(s_4,g) = 1 (since d is estimated only using successful trajectories) the cumulative distance of s_0 -> s_4 -> g is 3 (2 + 1) whereas the cumulative distance of s_0 -> s_1 -> s_2 -> g is 6 (3 + 2 + 1) and hence optimizing the cumulative distance does not lead to solving the pathological solution.

Any clarification will be greatly appreciated and my apologies if I had misunderstood your claims.

---

> ### Author Response · Authors · 2020-01-03
> **Author Reply for Clarifications in definition and Appendix A**
>
> Hey, thanks for the questions! Hopefully our answers below clarify the situation.
>
> > Can you please clarify whether the distance as defined in equation (2) computes the average number of steps to reach a state s_j from s_i for the first time or all visits? For example, in the sample trajectory s_1, s_2, s_3, s_4, s_5, s_3, s_6 is the distance between s_1 and s_3 2? or is it the average of all of {2, 3, 5}?
>
> The latter is true for DDL, i.e. for multiple visits between a pair of points, it will compute the average of the distances. In this specific example, the distance would be the average of $\{2, 3, 5\}$.
>
> > The proof presented in Appendix A assumes that all episodes reach the goal the state and the goal state is the terminal state.
>
> The assumption that all the episodes reach the goal state was originally used to guarantee that all the summations are finite, but is not needed in the current form because we have since included the discount factors in the summations. We’ll update this in the final version. Thanks for spotting that!
>
> > … hence optimizing the cumulative distance does not lead to solving the pathological solution.
>
> The problem here is with this assumption: “For example, the "value" of s_4 given by DDL is 1 whereas the true value of s_4 is infinity because S_T is absorbing.” This is actually backwards, i.e. the value of DDL policy is -infinity and the value of greedy policy is -1.
>
> The value given by DDL policy:
>     $V(s_4) = p * -d(s_4, g) + (1-p) * - (d(s_4, s_T) + d(s_T, g))) = p * -1 + (1-p) * -(1 + \infty) = -\infty$, when $p < 1$.
>
> The value of greedy policy:
>     $V(s_4) = -min(d(s_4,g), d(s_4, s_T) + d(s_T, g)) = -min(1, 1 + \infty) = -min(1, \infty) = -1$.
>
> Thus the greedy policy will prefer the path $s_0 \rightarrow s_4 \rightarrow g$, whereas the DDL policy will choose $s_0 \rightarrow s_1 \rightarrow s_2 \rightarrow g$.
>
> Hopefully that clarifies the case. We’re happy to elaborate if you have any further questions.

---

> > ### Public Comment · ~Academic_Anony_Mous1 · 2020-04-07
> > **Still unclear**
> >
> >
> >
> > I did not talk about any specific policy because my understanding of the so-called pathological MDP is that there is only action in s_4, there is only on policy in s_4 and taking any action leads to s_g and s_T with probabilities p and (1-p) respectively.
> >
> > The definition of dynamical distance given in equation (2) states that the \tau is sampled from the conditional distribution where s and s' occur in the same trajectory.
> >
> > In your response, you mention d(s_4, s_T) = \infty. Why? according to the definition in equation (2), taking the trajectories in which both s_4 and s_T occur, d(s_4, s_T) must always be 1. What am I missing here?
> >
> > In equation (4), the rewards are always computed using d(s_t, **g**). So, it is not clear why you introduced d(s_4, s_T) in the equations in your response. Can you kindly clarify the meaning of those equations?

---

> > > ### Author Response · Authors · 2020-04-25
> > > **Author Reply for Still unclear**
> > >
> > > Thanks a lot for pointing this out! You’re right, $d(s_4, s_T)$ should equal to $1$ in this example, whereas it's the $d(s_T, g)$ that should be $\infty$. The value of state $s_4$ thus ends up being $V(s_4) = -\infty$, because of the contribution of $d(s_T, g)$. I’ve corrected this in the answer above. Let us know if there's anything else unclear.

---

> > ### Public Comment · ~Academic_Anony_Mous1 · 2020-04-07
> > **Clarification in Appendix A**
> >
> > To my understanding, equation (8) in Appendix A only tells that the episodes are terminated as quickly as possible (set \gamma=1 for simplicity). So, if a (non-goal)terminal state is closer than the goal state, the agent will prefer the (non-goal)terminal state and not the desired goal state.
> >
> > Therefore, the agent does not necessarily prefer the non-risky path but a path to faster termination. Can you kindly clarify how is this avoiding non-risky states?

---

> > > ### Author Response · Authors · 2020-04-25
> > > **Author Reply for Clarification in Appendix A**
> > >
> > > We make an implicit assumption that there’s a large cost incurred for ending up in a non-goal terminal state, that is, for example, $d(s_T, g) = \infty$, for non-goal terminal state $s_T$ and desired goal $g$.

---

### Decision · Program_Chairs · 2019-12-19

**Decision:**

Accept (Poster)

**Comment:**

The authors present a method to learn the expected number of time steps to reach any given state from any other state in a reinforcement learning setting.  They show that these so-called dynamical distances can be used to increase learning efficiency by helping to shape reward.  After some initial discussion, the reviewers had concerns about the applicability of this method to continuing problems without a clear goal state, learning issues due to the dependence of distance estimates on policy (and vice versa), experimental thoroughness, and a variety of smaller technical issues.  While some of these were resolved, the largest outstanding issue is whether the proper comparisons were made to existing work other than DIAYN.  The authors appear to agree that additional baselines would benefit the paper, but are uncertain whether this can occur in time.  Nonetheless, after discussion the reviewers all appeared to agree on the merit of the core idea, though I strongly encourage the authors to address as many technical and baseline issues as possible before the camera ready deadline.  In summary, I recommend this paper for acceptance.